

# Two new species of Microlaimidae (Nematoda: Microlaimida) from the Continental Shelf off Northeastern Brazil (Atlantic Ocean) with emended diagnosis and dichotomous key

Alex Manoel, Patrícia F. Neres and Andre M. Esteves

Zoologia, Universidade Federal de Pernambuco, Recife, PE, Brazil

## ABSTRACT

New species of the genera *Spirobolbolaimus* and *Ixonema* (Nematoda: Microlaimidae) have been found in sediment samples collected in the South Atlantic, along the Continental Shelf break off Northeastern Brazil. Different to other *Spirobolbolaimus* species, *S. pernambucanus* **sp. nov.** possesses six outer labial setae and four cephalic setae with approximately the same length. *Ixonema gracieleae* **sp. nov.** differs from other species of *Ixonema* in having somatic setae on peduncles. This is the first time that new species of these taxa have been described for the Brazilian coast. An amendment of the diagnosis and a dichotomous key are proposed for both genera.

# INTRODUCTION

The family Microlaimidae (*Micoletzky, 1922*) currently comprises 13 genera and about 150 valid species (*Tchesunov, Jeong & Lee, 2021*; *Nemys, 2024*). Records of species belonging to this family are still scarce from the Brazilian coast and are almost all related to *Microlaimus De Man, 1880* (*Manoel, Neres & Esteves, 2024*). The oldest records in which species related to any genus placed in Microlaimidae were originally described for this location date back to the 1950s, when Gerlach described four species of *Microlaimus* for southeastern coast of Brazil (*Gerlach, 1956*; *Gerlach, 1957a*; *Gerlach, 1957b*). Three decades later, *Lima, Neres & Esteves (2022)* described three species of *Microlaimus* from the continental shelf off the Campos Basin, Southeastern Brazil. Recently, *Manoel, Neres & Esteves (2024)* described three new species of *Microlaimus* for the Continental Shelf break off Northeastern Brazil.

For other genera of Microlaimidae, such as *Spirobolbolaimus Soetaert & Vincx, 1988* and *Ixonema Lorenzen, 1971*, there are no records of species originally described for the South Atlantic. *Spirobolbolaimus* was placed in Microlaimidae based on male (two opposite and outstretched testes) and female (two outstretched ovaries) gonads (*Soetaert & Vincx, 1988*). *Spirobolbolaimus* is mainly characterized by having outer labial setae longer than the cephalic setae, multispiral amphidial fovea, postamphidial setae in six or eight longitudinal rows,

Corresponding author
Andre M. Esteves,
andresteves.ufpe@gmail.com

buccal cavity armed with protruding teeth and pharynx with anterior and posterior bulbs (*Soetaert & Vincx, 1988*; *Gourbault & Vincx, 1990*; *Shi & Xu, 2016*). The genus currently has three valid species: *S. bathyalis Soetaert & Vincx, 1988* (Mediterranean, Calvi Bay; 280–820 m depth); *S. boucherorum Gourbault & Vincx, 1990* (Caribbean Sea, Guadeloupe; 2 m depth) and *S. undulatus Shi & Xu, 2016* (China Sea, Nanji Islands; Dasha'ao sand beach).

*Ixonema* is the only representative of the family Microlaimidae that has three caudal glands with separate outlets, a feature considered rare in free-living marine nematodes (*Lorenzen, 1971*; *Lorenzen, 1994*). Often, representatives of this taxon are found with the cuticle covered by particles, such as algae, suggesting that these organisms are capable of carrying their own food (*Steyaert et al., 1999*). Additionally, this genus is characterized by a narrow and elongated anterior end, amphidial fovea far from the anterior end with a *corpus gelatum* projecting from the amphidial opening, a small mouth cavity and three minute teeth (*Tchesunov, 2014*). Only three species have been described for *Ixonema*: *I. sordidum Lorenzen, 1971* (North Sea, German Bight; sublittoral region, depth unspecified); *I. powelli Jensen, 1985* (Gulf of Mexico; 72 m depth) and *I. deleyi Muthumbi & Vincx, 1999* (Indian Ocean, Kenyan coast; 21–2,007 m depth).

Here we describe two new species, one from *Spirobolbolaimus* and another from *Ixonema*, found along the break of the continental shelf off Northeastern Brazil. Amendments to the diagnosis and dichotomous keys are proposed for both genera.

## MATERIAL AND METHODS

**Study area and sampling.** This information were previously described in *Manoel, Neres & Esteves (2024)*.

**Laboratory processing.** In the laboratory, sediment samples were sieved using a 500 μm mesh followed by a 45 μm mesh sieve which was used to retain the meiobenthic organisms. The samples remaining in the 45 μm mesh were extracted with colloidal silica (*Somerfield, Warwick & Moens, 2005*).

Nematoda were counted (and removed) under a stereomicroscope using a Dolffus plate. All individuals were transferred to a small glass container containing a solution with 99% formaldehyde (4%) + 1% glycerin (Solution 1 –*De Grisse, 1969*). The methodology of transferring each animal to glycerin was then applied, followed by diaphanization, according to the method described by *De Grisse (1969)*. The individuals were mounted permanently on glass slides, as an adaptation of the method described by *Cobb (1920)*. The genus was identified by using keys provided by *Warwick, Platt & Somerfield (1998)* and *Decraemer & Smol (2006)*. Species were identified through the comparison of their characteristics with those provided in the original descriptions. Drawings were made with the aid of an Olympus CX 31 optical microscope fitted with a drawing tube. Body measurements were taken using a mechanical map meter. The holotype and one paratype (female) of each species are deposited in the Nematoda Collection of the Museum of Oceanography Prof. Petronio Alves Coelho (MOUFPE), Brazil. Other paratypes are deposited in

the Meiofauna Laboratory, Zoology Department, Federal University of Pernambuco (NM LMZOO-UFPE).

The electronic version of this article in Portable Document Format (PDF) will represent a published study according to the International Commission on Zoological Nomenclature (ICZN), and hence the new names contained in the electronic version are effectively published under the Code from the electronic edition alone. This published research and the nomenclatural acts it contains have been registered in ZooBank, the online registration system for the ICZN. The ZooBank LSIDs (Life Science Identifiers) can be resolved and the associated information viewed through any standard web browser by appending the LSID to the prefix http://zoobank.org/. The LSID for this publication is: urn:lsid:zoobank.org: pub: urn:lsid:zoobank.org:pub:8B5A29A6-0EF8-454C-A62F-8C0B59390B83. The online version of this research is archived and available from the following digital repositories: PeerJ, PubMed Central and CLOCKSS.

# RESULTS

## Systematics

Class CHROMADOREA *Inglis, 1983*
Subclass CHROMADORIA *Pearse, 1942*
Order Microlaimida *Leduc, Verdon & Zhao, 2018*
Superfamily Microlaimoidea *Micoletzky, 1922*
Family Microlaimidae *Micoletzky, 1922*
Genus *Spirobolbolaimus Soetaert & Vincx, 1988*

**Diagnosis.** (Emended from *Shi & Xu, 2016*) Microlaimidae. Cuticle annulated. **Anterior sensilla in three circles: six inner labial setae papilliform; six stout outer labial setae (sometimes jointed); and four cephalic setae shorter or similar in length to the six outer labial setae**. **Buccal cavity large, armed with a well-developed dorsal tooth and a pair of ventrosublateral teeth. Additional lateral teeth may be present.** Amphidial fovea multispiral and ventrally wound, sclerotized. Postamphidial setae in six or eight longitudinal rows. Pharynx with anterior bulb. Oval posterior bulb present or absent. Females with two outstretched ovaries. Males with two opposed testes. Copulatory apparatus strongly sclerotized. Gubernaculum with or without apophysis. **Precloacal supplements (papilliform or small pores) present or absent**. **Oval, elongated or banana-shaped sperm cells. Tail conical**.

**Type species:** *Spirobolbolaimus bathyalis Soetaert & Vincx, 1988*.

## List of valid species of *Spirobolbolaimus* Soetaert & Vincx, 1988

*Spirobolbolaimus bathyalis* Soetaert & Vincx, 1988
*Spirobolbolaimus boucherorum* Gourbault & Vincx, 1990
*Spirobolbolaimus pernambucanus* **sp. nov.**
*Spirobolbolaimus undulatus* Shi & Xu, 2016

## Description of new species

*Spirobolbolaimus pernambucanus* **sp. nov.**
(Table 1; Figs. 1–4)

**Material studied**. Holotype male (MOUFPE 0022), paratype female (MOUFPE 0023), 1 male paratype (485 NM LMZOO-UFPE) and 2 female paratypes (486–487 NM LMZOO-UFPE).

**Type locality**. South Atlantic Ocean, Continental shelf off the State of Pernambuco, Brazil, (S08°16′17.10″ W34°39′34.80″), November 26, 2019, 52 m.

**Locality of paratypes**. Female paratypes: South Atlantic Ocean, Continental shelf off the State of Paraíba, Brazil (S07°25′15.48″ W34°29′16.56″), November 27, 2019, 47 m. Paratype male 1: South Atlantic Ocean, Continental shelf of the State of Pernambuco, Brazil (S08°16′17.10″ W34°39′34.80″), November 26, 2019, 52 m.

**Etymology**. Due to the location where the holotype was collected. *Pernambucanus* is the Latinized form of the term "pernambucano". In Brazil, "pernambucano" refers to something or someone originating from the state of Pernambuco.

**Holotype male**. Body cylindrical (1,716 μm long), slightly narrow anteriorly. Maximum body diameter corresponding to 1.2 times the head diameter. Head blunt, slightly set-off. Cuticle striated from the posterior edge of the amphidial fovea. Cuticular pores not observed. Anterior sensilla arranged in the 6+6+4 pattern: six inner labial papillae (2 μm long), six outer labial setae with a broad base (7 μm long) and four slender cephalic setae about the same length as the outer labial setae. Cephalic setae corresponding to 21% of head diameter. Amphidial fovea distinctly sclerotized, multispiral, ventrally wound, about 3.5 turns, occupying 36% of corresponding body diameter and located 0.4 times the head diameter of the anterior end. Buccal cavity with a strong dorsal tooth and two ventrosublateral teeth located at the same level. Cheilostoma possesses 12 rugae. Eight rows of cervical setae starting 11 μm behind the amphid fovea. Pharynx (220 μm) with anterior bulb surrounding the buccal cavity. Basal bulb is oval and not very prominent, occupying 67% of corresponding body diameter. Cardia embedded in intestine. Nerve ring situated at 55% of the pharynx length, from anterior end. Ventral gland and secretory-excretory pore not observed. Reproductive system with two opposed and outstretched testes, both to the left of the intestine. Elongated sperm cells. Spicules sclerotized with a capitulum. Gubernaculum plate-like with dorsal apophysis. Twelve pore-like precloacal supplements present and arranged at irregular intervals. The closest supplement to the cloaca is located at 24.5 μm and the farthest at 258.5 μm. A pair of precloacal ventrosublateral setae located

**Table 1  Morphometric data of *Spirobolbolaimus pernambucanus* sp. nov.**  The measurements are expressed in micrometers, or if noted, as a percentage or ratio. Not applicable (*); not available for measurement (-); a, b, c, c' = de Man's ratios (1880).

| *Spirobolbolaimus pernambucanus* sp. nov. | Holotype male | Male paratype (*n* = 1) | Females paratypes (*n* = 3) |
|---|---|---|---|
| Body length | 1,716 | 1,848 | 1,686–1,800 |
| Inner labial papillae length | 2 | – | 2 |
| Outer labial setae length | 7 | 7 | 7 |
| Cephalic setae length | 7 | 7 | 5.5–7 |
| Head diameter at level of the cephalic setae | 34 | 34 | 35–36 |
| Cephalic setae in relation to head diameter (%) | 21% | 21% | 15–19% |
| Distance from anterior end to amphidial fovea | 12 | 13 | 8–10.5 |
| Distance from anterior end to amphidial fovea in relation to head diameter | 0.4 | 0.4 | 0.2–0.3 |
| Amphidial fovea diameter (maximum width) | 12 | 12 | 11–12 |
| Body diameter at level of the amphidial fovea | 34 | 34 | 34.5–37 |
| % of the amphidial fovea diameter in relation to corresponding body diameter | 36% | 36% | 31–35% |
| Pharynx length | 220 | 223 | 226.5–241.5 |
| Position of nerve ring from anterior end | 120 | 133.5 | 125–138 |
| Nerve ring position in relation to pharynx length (%) | 55% | 60% | 55–56% |
| Pharyngeal bulb diameter | 26 | 30 | 28.5–31 |
| Body diameter at level of the pharyngeal bulb | 38 | 40 | 44–46.5 |
| % of basal bulb diameter in relation to corresponding body diameter | 67% | 75% | 64–66% |
| Maximum body diameter | 40 | 44.5 | 46.5–48 |
| Anal or cloacal body diameter | 31 | 36 | 32–33 |
| Tail length | 120 | 114 | 117–131 |
| Length of spicules along arc | 53 | 55 | * |
| Length of spicules along cord | 40 | 42 | * |
| Length of gubernaculum | 19.5 | 19 | * |
| Length of gubernaculum in relation to length of spicules along arc | 37% | 35% | * |
| Length of spicules along arc in relation to cloacal body diameter | 1.7 | 1.5 | * |
| Distance from anterior end to vulva | * | * | 867–984 |
| Position of vulva from anterior end (%) | * | * | 51–55% |
| Body diameter in vulva region | * | * | 46.5–48 |
| Anterior ovary length | * | * | 354–408 |
| Posterior ovary length | * | * | 252–288 |
| Reproductive system length | 927 | 1,104 | 642–666 |
| % of reproductive system in relation to body length | 54% | 60% | 36–39% |
| a | 43 | 42 | 35–39 |
| b | 8 | 8 | 7–7.5 |
| c | 14 | 16 | 13–14 |
| c' | 4 | 3 | 4 |

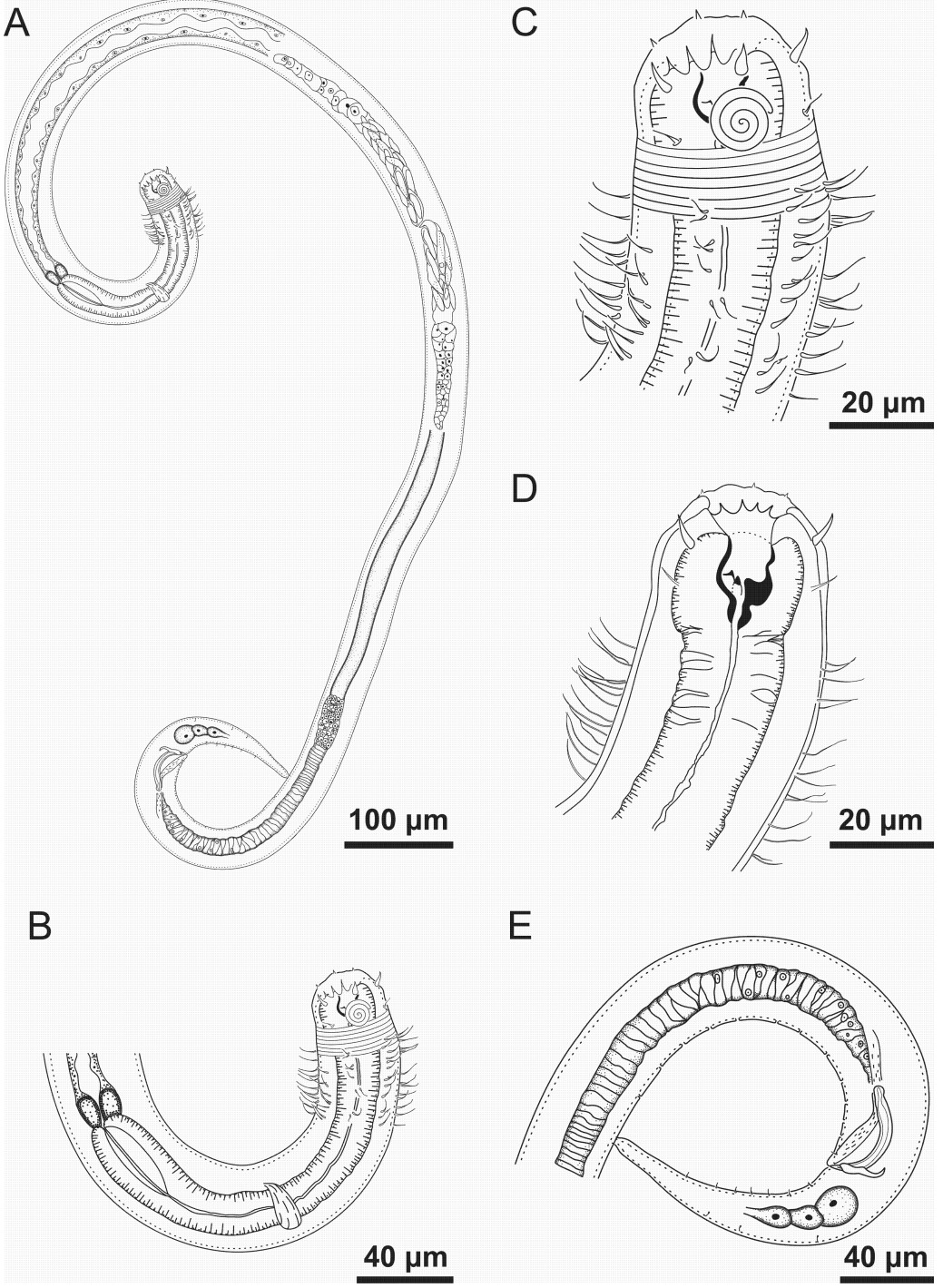

**Figure 1** *Spirobolbolaimus pernambucanus* **sp. nov. holotype male.** Holotype male: (A) overview; (B) anterior region; (C) anterior end (sensilla disposition, amphidial fovea and cervical setae); (D) anterior end (buccal cavity); (E) posterior region. Image source credit: Alex Manoel.

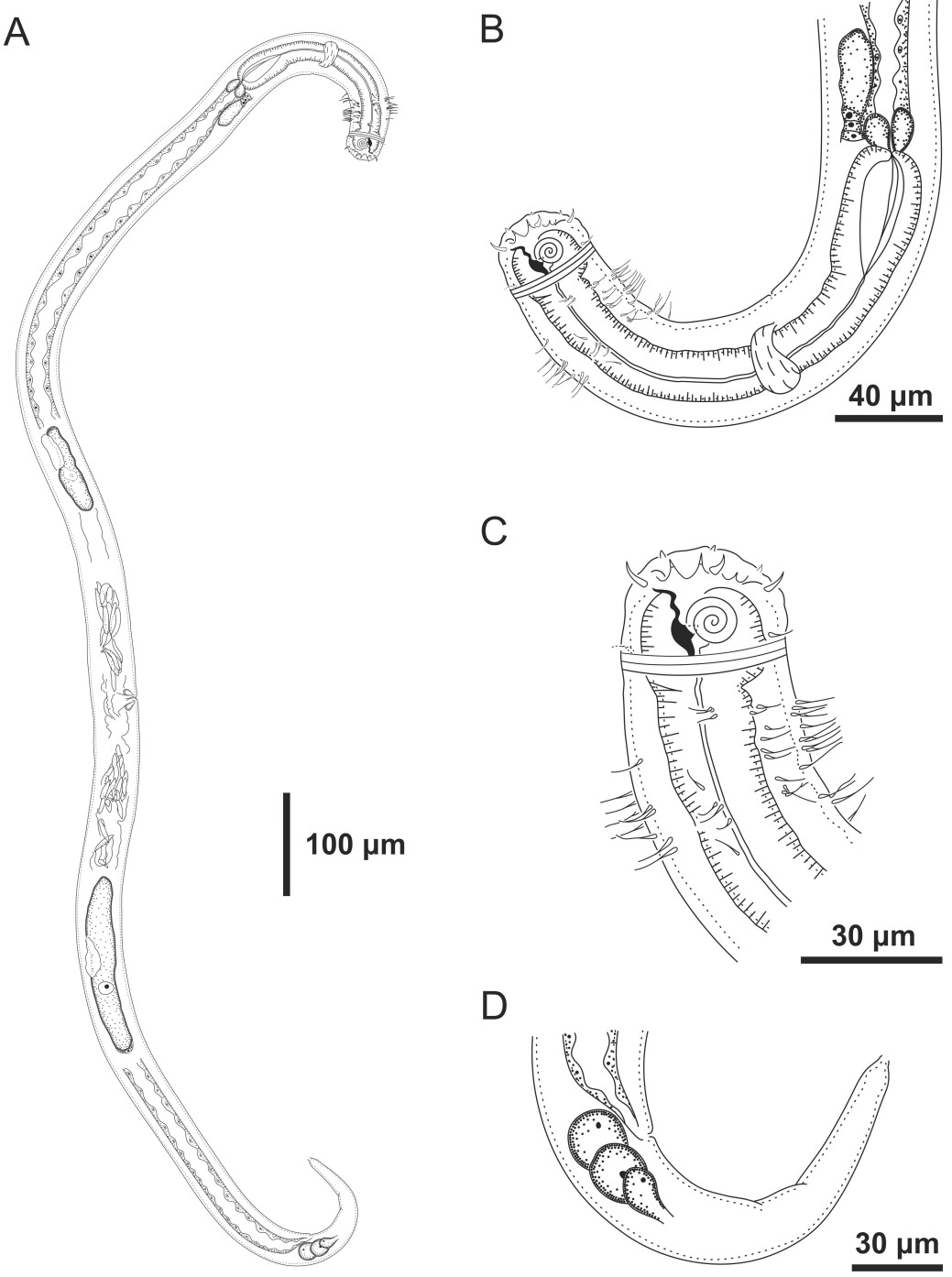

**Figure 2** *Spirobolbolaimus pernambucanus* **sp. nov. paratype female.** Paratype female: (A) overview; (B) anterior region; (C) anterior end (sensilla disposition, amphidial fovea and cervical setae); (D) tail. Image source credit: Alex Manoel.

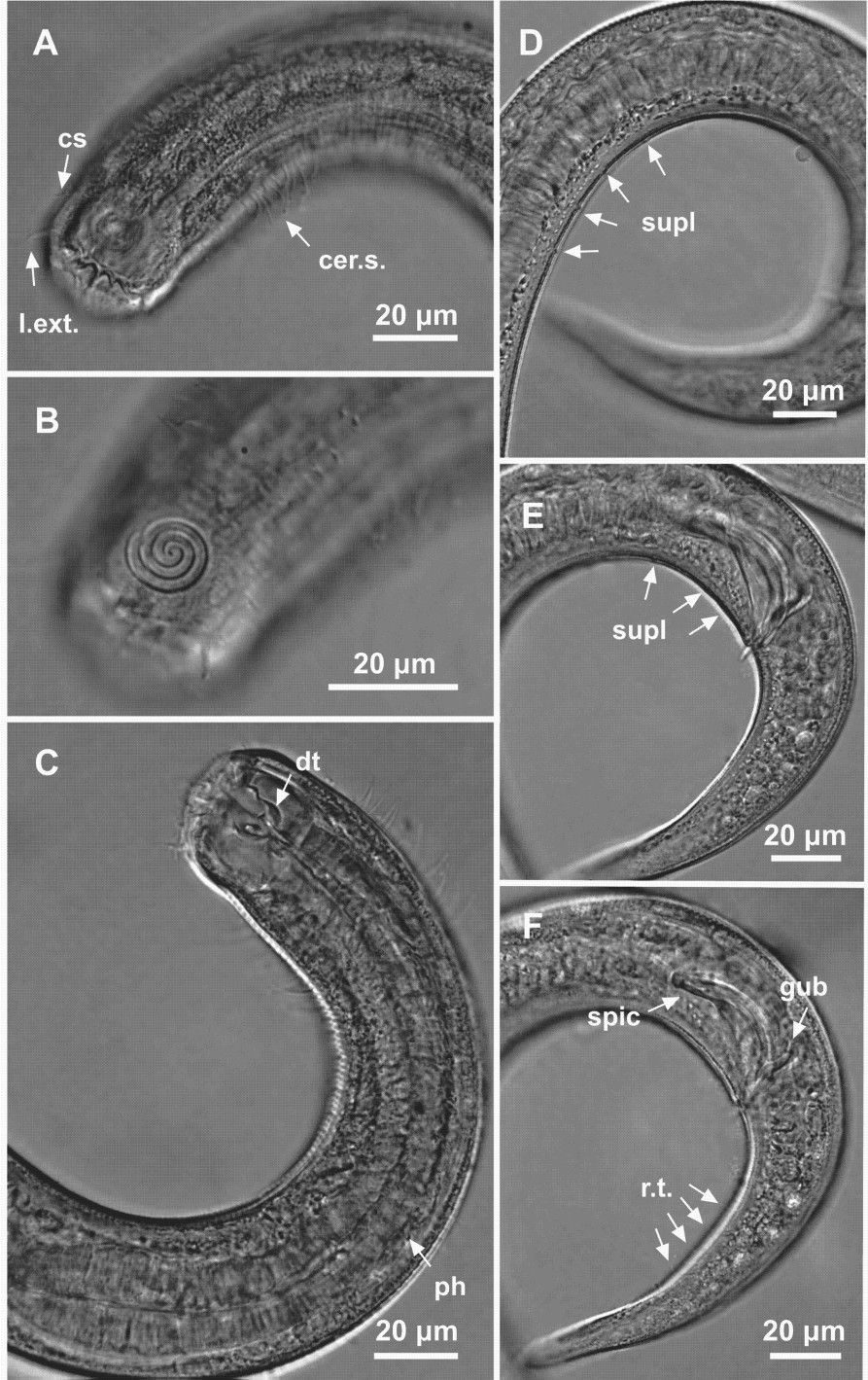

**Figure 3** *Spirobolbolaimus pernambucanus* **sp. nov.: holotype male.** Holotype male: (A) anterior region: arrows indicating outer labial setae (l.ext.), cephalic setae (cs) and cervical setae (cer. s.); (B) anterior region (amphidial fovea); (C) anterior end: arrows indicating the dorsal tooth (dt) and pharynx (ph); (D) posterior end: arrows indicating pore-like precloacal supplements farthest from the cloaca; (E) posterior end: arrows indicating pore-like precloacal supplements closer to the cloaca; (F) posterior end: arrow indicating the spicule (spic), gubernaculum (gub) and rows of setae on the tail (r.t.).

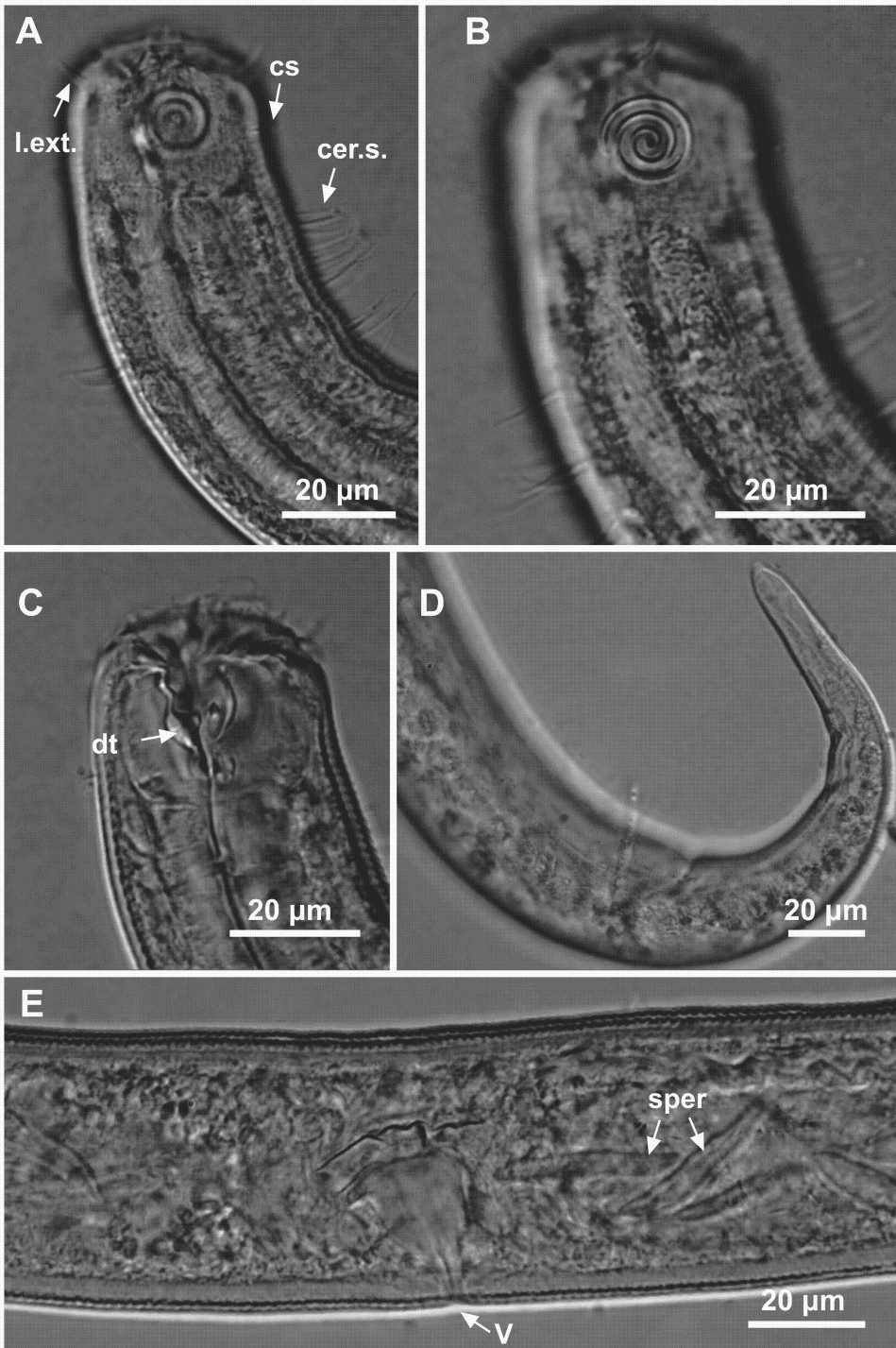

**Figure 4** *Spirobolbolaimus pernambucanus* **sp. nov. paratype female.** Paratype female: (A) anterior region: arrows indicating outer labial setae (l.ext.), cephalic setae (cs) and cervical setae (cer. s.); (B) anterior region (amphidial fovea); (C) anterior end: arrow indicating the dorsal tooth (dt); (D) tail; (E) vulva region: arrows indicating the vulva (V) and sperm (sper). Image source credit: Alex Manoel.

between the cloaca and the closest supplement to it. Three caudal glands. Tail conical (four times the cloacal body diameter) with two rows of short ventrosublateral setae.

**Paratype female**. Similar to male. Body length 1,752 µm, maximum diameter 46.5 µm (1.3 times the head diameter). Cuticle striated behind the posterior edge of the amphidial fovea. Anterior sensilla arrangement consisting of six inner labial papillae (2 µm long), six outer labial setae with a broad base (7 µm long) and four slender cephalic setae (6 µm long, corresponding to 17% of head diameter). Amphidial fovea sclerotized, multispiral, ventrally wound, about three turns, occupying 35% of corresponding body diameter and located 0.3 times the head diameter of the anterior end. Buccal cavity with a strong dorsal tooth and two ventrosublateral teeth. Pharynx similar to that of males. Basal bulb is oval and not very prominent (66% of corresponding body diameter). Nerve ring located at 56% of the pharynx length. Vulva situated at 55% of body length. Reproductive system didelphic-amphidelphic, with outstretched ovaries. Anterior gonad situated to right side of intestine and the posterior gonad to left side of intestine. Tail conical (131 µm long), about four times anal body diameter.

**Diagnosis**. *Spirobolbolaimus pernambucanus* **sp. nov.** it is characterized by its body length (1,716–1,848 µm). Cuticle striated. Head blunt, slightly set-off. Six outer labial setae with a broad base and four slender cephalic setae about the same length as the outer labial setae. Amphidial fovea sclerotized, multispiral, ventrally wound, about 3–3.5 turns, occupying 31–36% of the corresponding body diameter, located at about 0.2–0.4 times the head diameter. Eight rows of cervical setae. Buccal cavity with a strong dorsal tooth and two ventrosublateral teeth. Spicules sclerotized with a capitulum (1.5–1.7 times the cloacal body diameter) and gubernaculum plate-like with dorsal apophysis. Twelve pore-like precloacal supplements. Tail conical which corresponds to 3–4 cloacal or anal body diameter.

**Differential diagnosis.** *Spirobolbolaimus pernambucanus* **sp. nov.** shares the following features with *S. undulatus*: number of longitudinal rows of cervical setae (eight in both species), spicules length (53–55 µm in the new species and 51–55 µm in *S. undulatus*) and the ratio between spicules length and the cloacal body diameter (1.5–1.7 in *S. pernambucanus* **sp. nov.** and 1.3–1.5 in *S. undulatus*). Nevertheless, *S. pernambucanus* **sp. nov.** differs from *S. undulatus* in terms of body length (1,686–1,848 µm in the new species *vs* 2,035–2,558 µm in *S. undulatus*), the absence of jointed outer labial setae (*vs* outer labial setae jointed in *S. undulatus*), the presence of outer labial setae about the same length as the cephalic setae (*vs* outer labial setae longer than cephalic setae in *S. undulatus*) and the number of precloacal supplements (12 pore-like precloacal supplements in the new species *vs* 18–19 precloacal supplements in a series of mid-ventral elevations with pores on tops in *S. undulatus*). Furthermore, additional teeth are absent in *S. pernambucanus* **sp. nov.** (*vs* present in *S. undulatus*).

*Spirobolbolaimus pernambucanus* **sp. nov.** shares some features with adult specimens of *S. boucherorum*, such as: body length (1,686–1,848 µm in the new species *vs* 1,460–1,870 µm in *S. boucherorum*), amphidial fovea diameter (11–12 µm in *S. pernambucanus* **sp. nov.** and 10–12 µm in *S. boucherorum*) and de Man's ratio c (13–16 in the new species and 12–17 in *S. boucherorum*). However, *S. pernambucanus* **sp. nov.** differs from *S. boucherorum* in terms of the number of precloacal supplements (12 in the new species *vs* seven in *S. boucherorum*),

**Table 2  Comparison between the main features of valid species of *Spirobolbolaimus* *Soetaert & Vincx, 1988* .** The original descriptions were used to construct the table (Adult specimens only). The measurements are expressed in micrometers, or if noted, as a percentage or ratio. Absent (-); a, b, c, c' = de Man's ratios (1880); length of cephalic setae in relation to length of outer labial setae (cs. length/ols. length %); distance of amphidial fovea from anterior end in relation to head diameter (Amph/hd); length of spicules along arc in relation to cloacal body diameter (spic/c.b.d.); length of gubernaculum in relation to length of spicules along arc (gub/spic %).

| | *Spirobolbolaimus bathyalis* | *Spirobolbolaimus boucherorum* | *Spirobolbolaimus pernambucanus* sp. nov. | *Spirobolbolaimus undulatus* |
|---|---|---|---|---|
| L | 595–755 | 1,460–1,870 | 1,686–1,848 | 2,035–2,558 |
| a | 22.2–30.2 | 38.4–50 | 35–43 | 47–58 |
| b | 5–6.2 | 7.3–9.8 | 7–8 | 9–10.2 |
| c | 7.7–10.2 | 12.2–17.4 | 13–16 | 14–16.8 |
| c' | 3.5–4.5 | 2.9–3.6 | 3–4 | 3.8–4.4 |
| Outer labial setae | non-jointed | non-jointed | non-jointed | jointed |
| cs. length/ols. length % | 25–30% | 75% | 79–100% | 53–80% |
| Amph/hd | 0.3 | 0.25–0.3 | 0.2–0.4 | 0.3 |
| Number of turns of amphids | $4\frac{3}{4}$ turns | $2\frac{3}{4}$ turns | 3–3.5 turns | 3 turns |
| amph% | 55–65% | 29–35% | 31–36% | 39–43% |
| Rows of cervical setae | 6 | 8 | 8 | 8 |
| Precloacal supplements | – | 7 pore-like | 12 pore-like | 18–19 elevations with pores on tops |
| spic/c.b.d. | 1.5 | 1.7 | 1.5–1.7 | 1.3–1.5 |
| gub/spic % | 36–42% | 43% | 35–37% | 48–56% |

the length and shape of the gubernaculum (19–19.5 μm in *S. pernambucanus* *vs* 23–30 μm in *S. boucherorum*) and the presence of outer labial setae that are about the same length as the cephalic setae (*vs* outer labial setae longer than cephalic setae in *S. boucherorum*). A comparison of the main characters of all valid species of *Spirobolbolaimus* is presented in Table 2.

## Dichotomous identification key for valid species of *Spirobolbolaimus Soetaert & Vincx, 1988*

1. Six rows of cervical setae and precloacal supplements absent..…….. ..…..*S. bathyalis*
   –Eight rows of cervical setae and precloacal supplements present…………….….. 2
2. Outer labial setae jointed and additional lateral teeth present…………... *S. undulatus*
   –Outer labial setae non-jointed and additional lateral teeth absent..………………..... 3
3. Seven pore-like precloacal supplements and outer labial setae longer than cephalic setae……………….…..……………….............……..……………….  *S. boucherorum*
   –Twelve pore-like precloacal supplements and outer labial setae about the same length as the cephalic setae…………….....................……….. *S. pernambucanus* **sp. nov.**

## Genus *Ixonema Lorenzen, 1971*

**Diagnosis.** (Emended from *Tchesunov, 2014*) Microlaimidae. **Cuticle finely striated but can appear smooth** and may be covered with sediment particles. **Anterior sensilla**

arranged according to $6 + 6 + 4$ **pattern: six inner labial papillae; six outer labial papillae; and four cephalic setiform sensilla.** Anterior end narrowed and elongated. **Amphid fovea small circular or pocket-shaped, far posterior to the anterior end, with protruding rod-shaped *corpus gelatum* (not seen in some species, lost/broken or absent?).** Buccal cavity small, **armed with a dorsal tooth and a pair of ventrosublateral teeth (additional small teeth posterior to ventrosublateral may be present). Somatic setae stout or on peduncles, sometimes jointed, arranged in rows along the body. Males with two opposite and outstretched testicles. Precloacal supplements in the form of jointed or non-jointed setae may be present. Ventral supplements in the form of jointed setae located just behind the pharynx may be present. Gubernaculum present or absent, when present without apophyses. Female didelphic-amphidelphic, with outstretched ovaries. Tail conical.** Each caudal gland with its own outlet on the tail tip.

**Type species:** *Ixonema sordidum* *Lorenzen, 1971*

## List of valid species of *Ixonema Lorenzen, 1971*

*Ixonema deleyi* *Muthumbi & Vincx, 1999*
*Ixonema gracieleae* **sp. nov.**
*Ixonema powelli* *Jensen, 1985*
*Ixonema sordidum* *Lorenzen, 1971*

## Description of new species

*Ixonema gracieleae* **sp. nov.**
(Table 3; Figs. 5–9)

**Material studied**. Holotype male (MOUFPE 0024), paratype female (MOUFPE 0025), 3 male paratypes (488–490 NM LMZOO-UFPE) and 2 female paratypes (491–492 NM LMZOO-UFPE).

**Type locality**. South Atlantic Ocean, Continental shelf off the State of Alagoas (S09°39′14.52″ W35°15′21.66″), November 25, 2019, 50 m. The paratypes were found in the same locality.

**Etymology**. The specific epithet is a tribute to Graciele Mariza dos Santos Alves, wife of the first author.

**Holotype male**. Body cylindrical, short and plump, attenuating on both ends and thick in the middle (586.5 $\mu$m long). Maximum body diameter corresponding to 4.1 times the head diameter. Cuticle finely striated, the striations are so delicate that they are difficult to visualize (striations are most visible on the tail). Anterior sensilla arranged in the $6+6+4$ pattern: six inner labial papillae, six outer labial papillae and four cephalic setae (6 $\mu$m long). Cephalic setae corresponding to 74% of head diameter. Amphidial fovea small and circular, occupying 25% of corresponding body diameter and located far posterior to the anterior end (four times the head diameter of the anterior end). Protruding *Corpus*

**Table 3  Morphometric data of *Ixonema gracieleae* sp. nov.** The measurements are expressed in micrometers, or if noted, as a percentage or ratio. Not applicable (*); a, b, c, c' = de Man's ratios (1880).

| *Ixonema gracieleae* sp. nov. | Holotype male | Males paratypes ($n = 3$) | Females paratypes ($n = 3$) |
|---|---|---|---|
| Body length | 586.5 | 562.5–630 | 553.5–667.5 |
| Outer labial setae length | <2 | <2 | <2 |
| Cephalic setae length | 6 | 5–7 | 6–7 |
| Head diameter at level of the cephalic setae | 8 | 8–9 | 8–9 |
| Cephalic setae in relation to head diameter (%) | 74% | 67–82% | 71–88% |
| Distance from anterior end to amphidial fovea | 31 | 31.5–34 | 29–32.5 |
| Distance from anterior end to amphidial fovea in relation to head diameter | 4 | 3.7–4.1 | 3.2–3.9 |
| Amphidial fovea diameter (maximum width) | 3 | 3 | 2.5–3 |
| Body diameter at level of the amphidial fovea | 13 | 14 | 13–14 |
| % of the amphidial fovea diameter in relation to corresponding body diameter | 25% | 21–22% | 19%–23% |
| Pharynx length | 104 | 99–104 | 96–102 |
| Position of nerve ring from anterior end | 63 | 68–71 | 64–67 |
| Nerve ring position in relation to pharynx length (%) | 60% | 68% | 65–67% |
| Pharyngeal bulb diameter | 18 | 17.5–19 | 19–19.5 |
| Body diameter at level of the pharyngeal bulb | 26 | 24–26 | 25–26 |
| % of basal bulb diameter in relation to corresponding body diameter | 68% | 70–78% | 74–76% |
| Maximum body diameter | 32 | 31–32 | 35–36 |
| Anal or cloacal body diameter | 21 | 20 | 20–20.5 |
| Tail length | 73 | 65.5–69 | 64–67 |
| Length of spicules along arc | 39 | 33–41 | * |
| Length of spicules along cord | 33 | 30–32 | * |
| Length of gubernaculum | 11.5 | 13–14.5 | * |
| Length of gubernaculum in relation to length of spicules along arc | 29% | 35–38% | * |
| Length of spicules along arc in relation to cloacal body diameter | 1.9 | 1.7–2 | * |
| Distance from anterior end to vulva | * | * | 261–411 |
| Position of vulva from anterior end (%) | * | * | 47%–62% |
| Body diameter in vulva region | * | * | 35–36 |
| Anterior ovary length | * | * | 52–69 |
| Posterior ovary length | * | * | 42–69 |
| Reproductive system length | 431 | 425–479.5 | 105–138 |
| % of reproductive system in relation to body length | 73% | 67%–82% | 16–25% |
| a | 18 | 18–20 | 15–19 |
| b | 6 | 6 | 5–7 |
| c | 8 | 9 | 8–10 |
| c' | 3.5 | 3–3.5 | 3 |

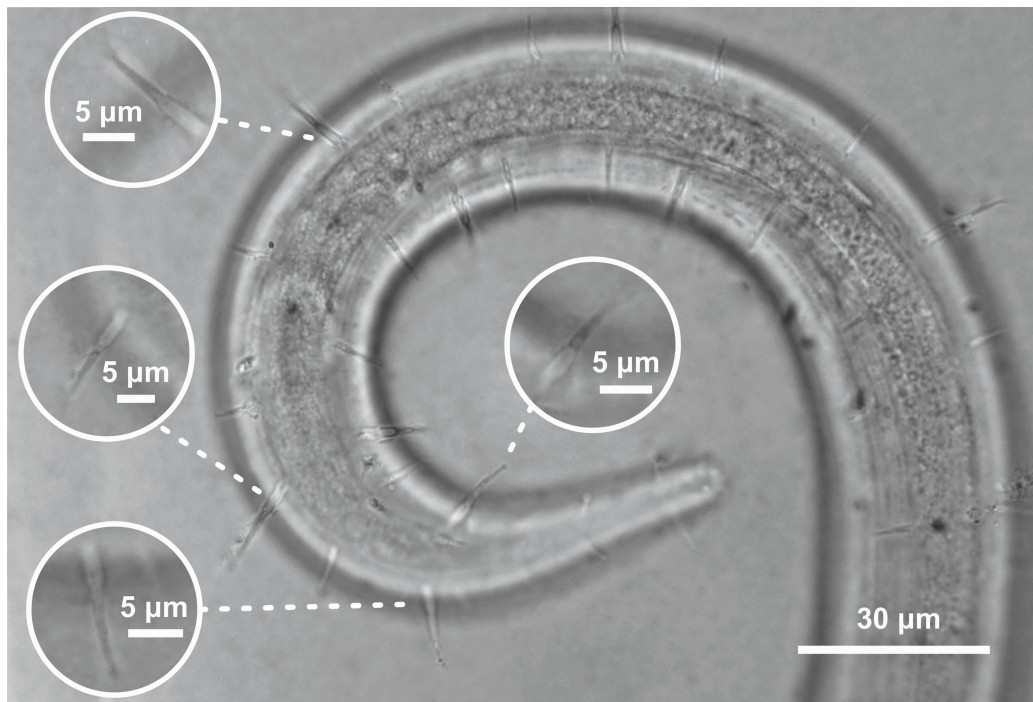

**Figure 5** ***Ixonema gracieleae* sp. nov. holotype male.** Setae on peduncles distributed along the body. Image source credit: Alex Manoel.

*gelatum* not observed. Buccal cavity with a dorsal tooth and two small ventrosublateral teeth. Cheilostoma rugae indiscernible under a light microscope. Four sublateral rows of stout cervical setae located between the posterior region of the cephalic setae and the base of the amphidial fovea. The first row is 24 µm from the anterior end and the second is at the same height of the amphidial fovea base, 31 µm from the anterior end. Six rows of cervical setae on peduncles located between the posterior region of the amphidial fovea and close to the bulb base: four sublateral rows where relatively smaller setae (4–7 µm) and long setae (10–16 µm) alternate; two lateral rows (about 4 µm). The first row is about 11 µm from the base of the amphidial fovea. After the cervical region, the somatic setae are distributed in two rows where the alternation between two smaller setae (about 5–9 µm) and a larger one (12–19 µm) usually occurs. Pharynx (104 µm) with prominent terminal bulb, occupying 73% of corresponding body diameter. Cardia embedded in intestine. Nerve ring situated at 60% of the pharynx length, from anterior end. Ventral gland and secretory-excretory pore not observed. Reproductive system with two opposed and outstretched testes on right side of intestine. Spicules curved (39 µm), about two times the cloacal body diameter. Gubernaculum slender and without apophysis. Four setiform ventral supplements: three precloacal supplements plus one located just behind the pharynx. The two setae closest to the cloaca (10.5 µm and 40 µm anterior to the cloaca, respectively) are jointed, measuring 7 µm and 6 µm respectively; the third precloacal seta is smaller (about 2 µm length), apparently non-jointed in light microscopy and far from the cloaca (62 µm). The setae

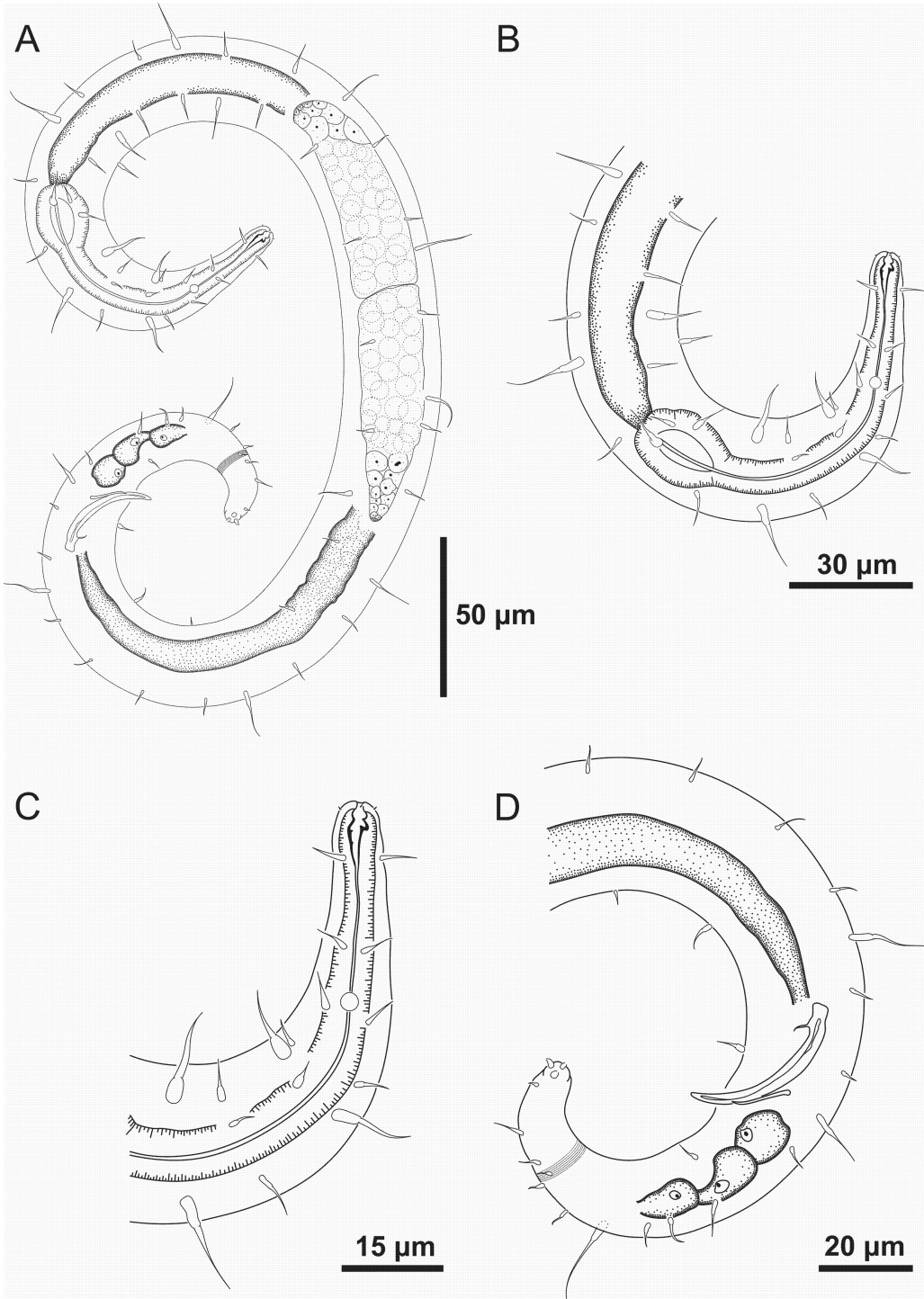

**Figure 6** *Ixonema gracieleae* **sp. nov. holotype male.** Holotype male: (A) overview; (B) anterior region; (C) anterior end; (D) posterior region. Image source credit: Alex Manoel.

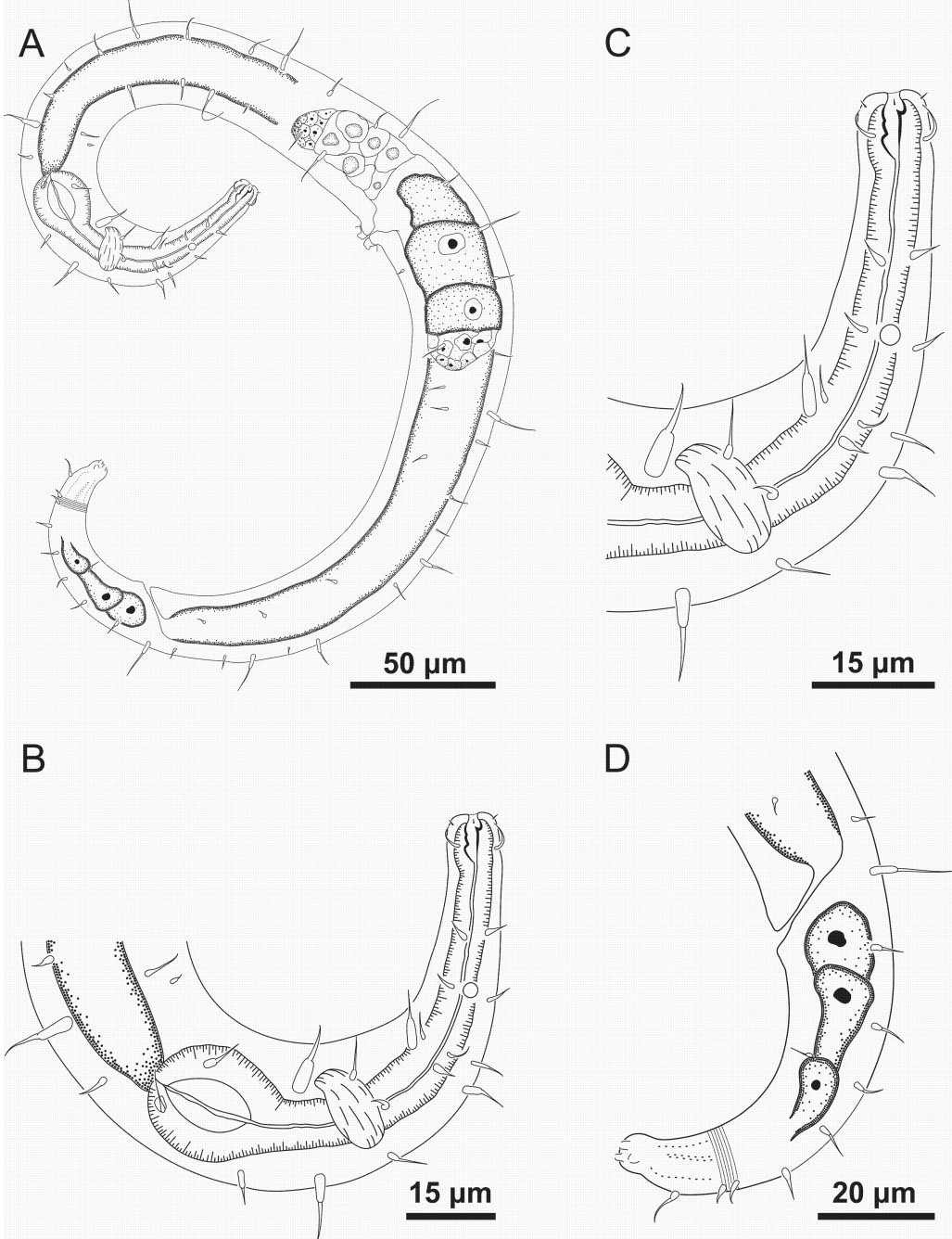

**Figure 7  *Ixonema gracieleae* sp. nov.: paratype female.** Paratype female: (A) overview; (B) anterior region; (C) anterior end; (D) tail. Image source credit: Alex Manoel.

located just behind the pharynx (about 7 µm length) are jointed (morphologically similar to the two setae closest to the cloaca) and are located 84 µm from the anterior end. Tail conical (73 µm long), with a blunt tip where the three caudal glands open through separate outlets into papilla-like extensions.

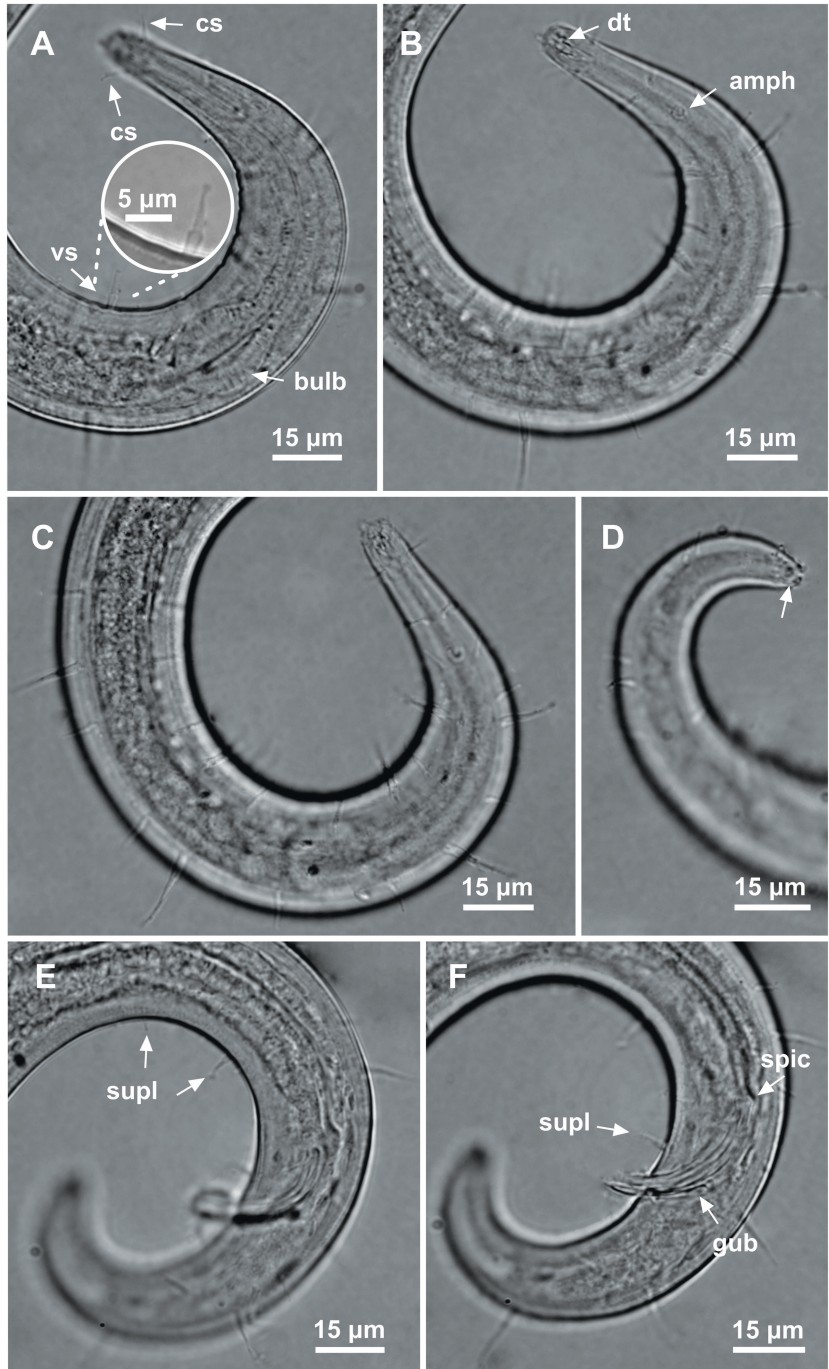

**Figure 8** *Ixonema gracieleae* **sp. nov. holotype male.** Holotype male: (A) anterior region: arrows indicating cephalic setae (cs), a single ventral seta in the anterior region (vs) and pharyngeal bulb (bulb); (B) anterior region: arrows indicating the dorsal tooth (dt) and amphidial fovea (amph); (C) anterior end (somatic setae); (D) tail (arrow indicating three separate tail tip outlets); (E) posterior end: arrows indicating precloacal supplements (supl); (F) posterior end: arrows indicating precloacal supplement (supl), spicule (spic) and gubernaculum (gub). Image source credit: Alex Manoel.

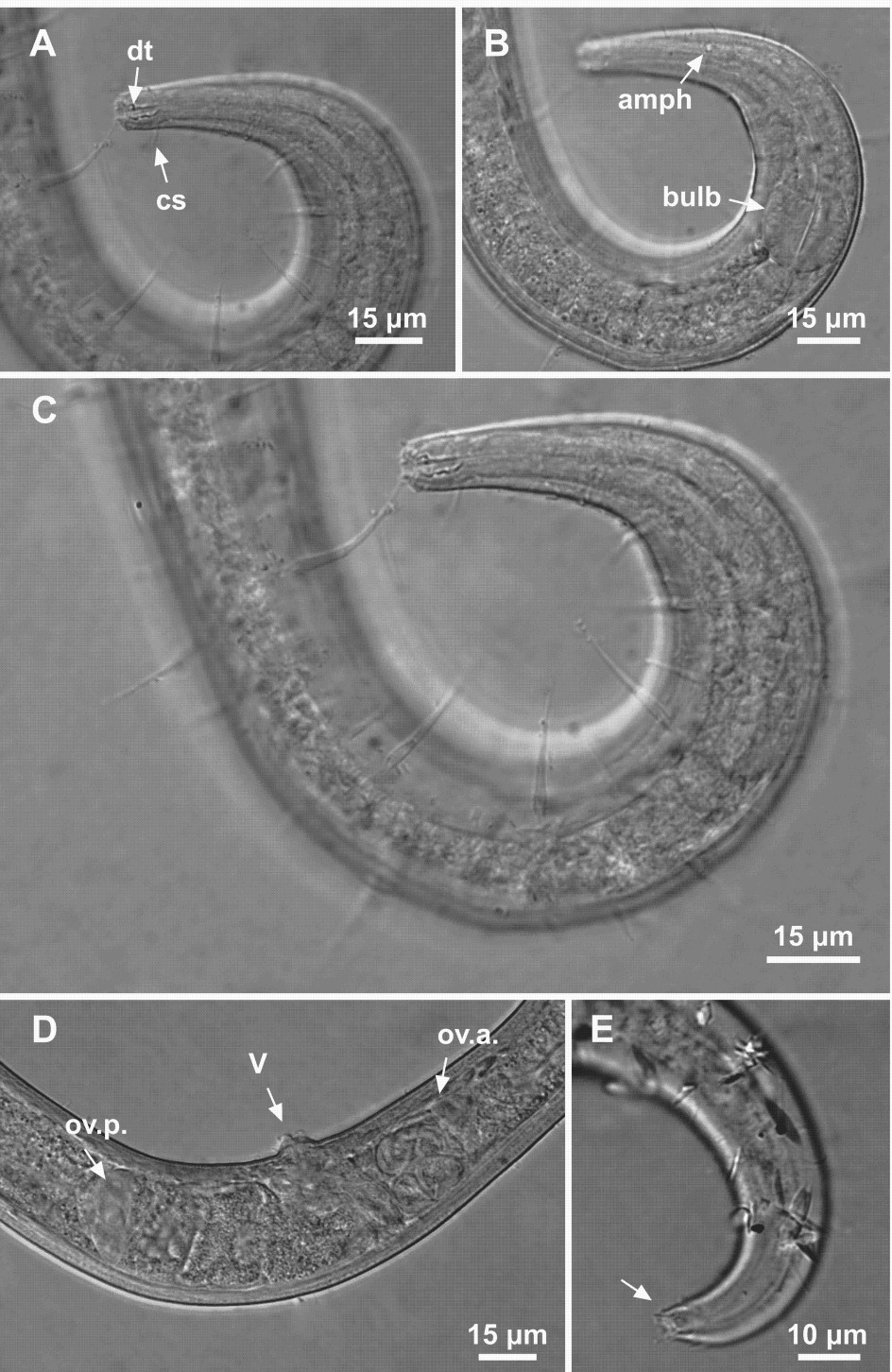

**Figure 9** *Ixonema gracieleae* **sp. nov. paratype female 1 and paratype female 2.** Paratype female 1: (A) anterior region: arrows indicating cephalic setae (cs) and the dorsal tooth (dt); (B) anterior region: arrows indicating amphidial fovea (amph) and pharyngeal bulb (bulb); (C) anterior end (somatic setae); (D) reproductive system: arrows indicating the anterior ovary (ov.a.), posterior ovary (ov.p.) and the vulva (V). Paratype female 2: (E) tail (arrow indicating three separate tail tip outlets). Image source credit: Alex Manoel.

**Paratype female**. Similar to male. Body measuring 667.5 μm in length, maximum diameter 36 μm (4.3 times the head diameter). Cuticle finely striated, the striations are so delicate that they are difficult to visualize. Anterior sensilla arranged in the 6 + 6 + 4 pattern: six inner labial papillae, six outer labial papillae and four cephalic setae (7 μm long, corresponding to 88% of head diameter). Amphidial fovea small and circular, occupying 21% of corresponding body diameter and located far posterior to the anterior end (3.9 times the head diameter of the anterior end). Protruding *corpus gelatum* not observed. Morphologies and distribution patterns of cervical and somatic setae are similar to the male. Jointed ventral seta located just behind the pharynx absent. Buccal cavity, teeth and pharynx similar to that of male. Bulb occupying 74% of corresponding body diameter. Nerve ring situated at 67% of the pharynx length, from anterior end. Vulva located 411 μm from anterior end, at 62% of body length. A pair of ventral papillae surround the vulva. Reproductive system didelphic-amphidelphic, with outstretched ovaries. Anterior and posterior ovary to right side of intestine. Tail conical (67 μm long), similar to the male.

**Diagnosis**. *Ixonema gracieleae* **sp. nov.** it is characterized by its finely striated cuticle. Amphidial fovea small and circular located far posterior to the anterior end. Protruding *corpus gelatum* not observed. Rows of cervical and somatic setae on peduncles along the body. Reproductive system with two opposed and outstretched testes in the males and didelphic-amphidelphic, with outstretched ovaries in the females. Spicules curved and gubernaculum without apophysis. Four setiform ventral supplements: three precloacal setae (the two closest to the cloaca are jointed and the furthest apparently non-jointed) and a jointed ventral seta located just behind the pharynx. Tail conical with a blunt tip where the three caudal glands open through separate outlets into papilla-like extensions.

**Differential diagnosis.** *Ixonema gracieleae* **sp. nov.** resembles the monospecific genus *Bathynox* (*Bussau, 1993*) *Bussau & Vopel, 1999* due to the presence of the setae on peduncles distributed along the whole body. However, *I. gracieleae* **sp. nov.** differs from *Bathynox* in having three caudal glands opening through separate outlets in papilla-like extensions (*vs* three glands opening in a single outlet in *Bathynox*). Furthermore, the male reproductive system is different between the genera: male diorchic in *I. gracieleae* **sp. nov.** *vs* male monorchic in *Bathynox* and the gubernaculum lacks apophysis in *I. gracieleae* **sp. nov.** *vs* the presence of dorsal apophysis in *Bathynox*.

The presence of setae on peduncles in *I. gracieleae* **sp. nov.** differentiates it from other species of the genus. The new species shares the presence of a single jointed ventral seta located just behind the pharynx with *I. powelli*, as well as jointed precloacal setae in the males. Nonetheless, *I. gracieleae* **sp. nov.** differs from *I. powelli* in terms of the shape of the amphidial fovea (circular in *I. gracieleae* **sp. nov.** *vs* pocket-like in *I. powelli*), the distribution pattern and number of rows of cervical setae (6 rows of cervical setae in *I. gracieleae* **sp. nov.** *vs* 4 rows of cervical setae *I. powelli*) and gubernaculum length (11.5–14.5 μm in *I. gracieleae* **sp. nov.** *vs* about 6 μm in *I. powelli*).

*Ixonema gracieleae* **sp. nov.** differs from *I. sordidum* in having long somatic setae alternating with short somatic setae along the body and due to the presence of precloacal supplements, these characteristics are absent in the mentioned species. *I. gracieleae* **sp. nov.** differs from *I. deleyi* in terms of the presence of the gubernaculum (*vs* absent in *I. deleyi*)

**Table 4  Comparison between the main features of valid species of *Ixonema* *Lorenzen, 1971*.** The original descriptions were used to construct the table. The measurements are expressed in micrometers, or if noted, as a percentage or ratio. Present (+) or absent (-); a, b, c, c' = de Man's ratios (1880); corpus gelatum (*cg.*); distance of amphidial fovea from anterior end in relation to head diameter (Amph/hd); length of spicules along arc in relation to cloacal body diameter (spic/c.b.d.); length of gubernaculum in relation to length of spicules along arc (gub/spic %); not applicable (x).

| | *Ixonema deleyi* | *Ixonema gracieleae* sp. nov. | *Ixonema powelli* | *Ixonema sordidum* |
|---|---|---|---|---|
| L | 245–405 | 553.5–667.5 | 640–900 | 530 |
| a | 12.3–22.5 | 15–19 | 13–23 | 16–20 |
| b | 3.6–4.9 | 5–7 | 5.9–7 | 5.1–5.3 |
| c | 7.3–9.2 | 8–10 | 7.3–10.6 | 8.2–8.8 |
| c' | 2.5–4.5 | 3–3.5 | 3.5 | 3 |
| Amphidial fovea | circular (rod-like *cg.*) | circular (*cg.* not seen or absent) | pocket-shaped (*cg.* not seen or absent) | circular (rod-like *cg.*) |
| Amph/hd | 4.8–6.2 | 3.2–4 | 4–5 | 3.6–3.8 |
| Somatic setae | stout | on peduncles | jointed | stout |
| Supplements | 1 non-jointed precloacal setae | 4 jointed setae* (1 just behind pharynx +3 precloacal) | 3 jointed setae (1 just behind pharynx +2 precloacal) | - |
| spic/c.b.d. | 2 | 1.7–2 | 1.4–1.8 | 1.4 |
| Gubernaculum | - | + | + | + |
| gub/spic % | x | 29–38% | 14% | 44% |

**Notes.**
*Precloacal seta furthest from the cloaca apparently non-jointed using light microscopy in *Ixonema gracieleae* **sp. nov.**

and the number and morphology of supplements (jointed ventral seta located just behind the pharynx + two jointed setae closer to the cloaca and a smaller non-jointed seta further away in *I. gracieleae* **sp. nov.** *vs* a single non-jointed ventral precloacal seta in *I. deleyi*). A comparison of the main characters of all valid species of *Ixonema* is presented in Table 4.

### Dichotomous identification key for valid species of *Ixonema Lorenzen, 1971*

1. Body length less than 500 µm and gubernaculum absent……………………..…*I. deleyi*
–Body length longer than 500 µm and gubernaculum present…………………………2
2. Somatic setae stout and precloacal supplements absent…………………….. *I. sordidum*
–Somatic setae on peduncles or jointed and precloacal supplements present……….. 3
3. Amphidial fovea circular and somatic setae on peduncles … *I. gracieleae* **sp. nov.**
–Amphidial fovea pocket-shaped and somatic setae jointed ………….....…..... *I. powelli*

### DISCUSSION

Although there are no records of species originally described for the South Atlantic, the occurrence of *Spirobolbolaimus* and *Ixonema* was previously reported for this region. *Spirobolbolaimus* was found in the sublittoral of Pedra do Xaréu Beach, Pernambuco, Northeastern Brazil (*Rocha et al., 2006*). This taxon was also found in sediment samples from the Grussaí canyon and a point adjacent to it in the Campos Basin, Southeastern

Brazil (*Silva, 2012*). *Ixonema* was identified in samples from the Campos Basin Slope, Southeastern Brazil (*Moura, 2013*).

*Spirobolbolaimus pernambucanus* **sp. nov.** is the first species of the genus described for the South Atlantic. In all previously described species, the outer labial setae are longer than the cephalic setae. However, the outer labial setae of *S. pernambucanus* **sp. nov.** are similar in length to the cephalic setae. We included this feature in the diagnosis of the genus. Based on the described species, we added the following characteristics to the diagnosis of the genus: the morphology of the buccal cavity and teeth that usually occur in species; the types of precloacal supplements; sperm cells and tail shape; occurrence of jointed outer labial setae. Jointed labial setae, as seen in *S. undulatus*, may be present in more genera or species than mentioned in the descriptions, since in some cases this characteristic may have been overlooked (*Lorenzen, 1994*).

*Ixonema gracieleae* **sp. nov.** can be easily confused and classified as belonging to *Bathynox*, mainly due to the presence of rows setae on peduncles distributed along the body. However, *I. gracieleae* **sp. nov.** presents a combination of differential characteristics that typically only occur in representatives of *Ixonema*: three caudal glands open through separate outlets into papilla-like extensions, two opposite and outstretched testes and gubernaculum without apophysis (*Lorenzen, 1971*; *Jensen, 1985*; *Muthumbi & Vincx, 1999*). Therefore, the presence of such characteristics invalidates the hypothesis of including this species in another genus and reinforces its taxonomic position. In the Microlaimidae family, the presence of caudal glands with independent outlets is unique for *Ixonema*, and this feature is considered phylogenetically primitive (*Lorenzen, 1971*). *Ixonema gracieleae* **sp. nov.** is the first species of the genus described from the South Atlantic. The description of this new species strongly contributes to the knowledge and the variability of this genus.

Among the three valid species of *Ixonema*, it was possible to observe variations in some important characteristics for the identification of the genus that were absent in the last diagnosis of the genus provided by *Tchesunov (2014)*. The cuticle of the species *I. sordidum* and *I. powelli* was described as smooth (*Lorenzen, 1971*; *Jensen, 1985*). In both species, the cuticle was covered by a thin layer of particles, except in a *I. powelli* male, where it was possible to visualize subcuticular striae in the most anterior region of the pharynx. Electron microscopy analysis allowed the description of a very finely striated cuticle for *I. deleyi* (*Muthumbi & Vincx, 1999*). *Tchesunov, Jeong & Lee (2021)* included this variation in a comparative table between the genera of Microlaimidae but did not provide a complete diagnosis of *Ixonema*. In the new species, the cuticle is similar to that described for *I. deleyi*, and it is possible to visualize very fine striations on the tail of the analyzed specimens, even using optical microscopy. Therefore, the cuticle of *Ixonema* is very finely striated, but can also appear smooth, especially when using light microscopy.

The amphidial fovea can vary from circular (*I. sordidum*, *I. deleyi* and *I. gracieleae* **sp. nov.**) to pocket-like (*I. powelli*). A protruding rod-shaped *corpus gelatum* is present in *I. sordidum* and *I. deleyi*. This characteristic was not observed in *I. powelli* and *I. gracieleae* **sp. nov.** When establishing the genus, *Lorenzen (1971)* argued that the walls of the gelatinous rods do not appear to be delicate structures. However, although *I. deleyi* presents a protruding *corpus gelatum*, in the electron microscopy analyzes provided in the original

description of this species, it is possible to observe that this structure seems to have been lost during specimen preparation (see in *Muthumbi & Vincx, 1999*-figure 7). This observation is not in line with Lorenzen's assumption (non-delicate structures). Therefore, we do not know for sure if this structure is absent in *I. powelli* and *I. gracieleae* **sp. nov.** or if it was lost/broken during sample processing and organism preparation, which commonly occurs with other structures such as the tail and setae.

The gubernaculum is present and lacks apophyses in the species *I. gracieleae* **sp. nov.**, *I. sordidum* and *I. powelli*. However, this structure is absent in *I. deleyi*. Supplements in the form of jointed setae in a ventral position are present just behind the pharynx and in front of the cloaca in *I. powelli* and *I. gracieleae* **sp. nov.** A non-jointed ventral seta located anterior to the cloacal opening is present in *I. deleyi*. Due to the similarity with *I. powelli* and *I. garacielea* **sp. nov.** in relation to the position at which the seta is located, we will consider it as a precloacal supplement. Jointed somatic setae may be present (*I. powelli*). The occurrence of somatic setae on peduncles described for the new species is unprecedented for the genus. The variation of the characters discussed above, in addition to the new features found in the new species, were included in the diagnosis of the genus.

The present study increases the number of Microlaimidae species originally described from sediment samples collected in the Brazilian coast. These results demonstrate that a great effort is still required in order to fully understand the real richness of the Microlaimidae assemblage present in marine sediments of continental margins, such as the Brazilian coast.

# ACKNOWLEDGEMENTS

We are very grateful for Dra. Tania Nara Bezerra (UGent, Belgium) and two anonymous reviewers for all comments/suggestions to improve this manuscript.

## Funding
The Brazilian navy provided logistical support for the scientific cruise aboard the R/V Vital de Oliveira. Alex Manoel has a FACEPE graduate scholarship (IBPG-1516-2.00/21). The Coordenação de Aperfeiçoamento de Pessoal de Nível Superior - Brasil (CAPES) - Finance Code 001 funded the APC and only the APC for this article. The funders had no role in study design, data collection and analysis, decision to publish, or preparation of the manuscript.

## Grant Disclosures
The following grant information was disclosed by the authors:
The scientific cruise aboard the R/V Vital de Oliveira.
FACEPE graduate scholarship: IBPG-1516-2.00/21.
Coordenação de Aperfeiçoamento de Pessoal de Nível Superior - Brasil (CAPES): Finance Code 001.

## Competing Interests

The authors declare there are no competing interests.

## Author Contributions

- Alex Manoel performed the experiments, analyzed the data, prepared figures and/or tables, authored or reviewed drafts of the article, and approved the final draft.
- Patrícia F. Neres performed the experiments, analyzed the data, authored or reviewed drafts of the article, and approved the final draft.
- Andre M. Esteves conceived and designed the experiments, performed the experiments, analyzed the data, authored or reviewed drafts of the article, and approved the final draft.

## Field Study Permissions

The following information was supplied relating to field study approvals (i.e., approving body and any reference numbers):

This is a study carried out by the Brazilian government. This study was carried out by BRAZILIAN NAVY and licensed of ENVIRONMENTAL MINISTRY OF BRAZIL

## Data Availability

The data are available in the Tables.

## New Species Registration

The following information was supplied regarding the registration of a newly described species:

Publication LSID: urn:lsid:zoobank.org:pub:8B5A29A6-0EF8-454C-A62F-8C0B59390B83

Spirobolbolaimus genus LSID: urn:lsid:zoobank.org:act:547D206A-82E4-4894-ACC2-B1C54C98AF90

Spirobolbolaimus pernambucanus species LSID: urn:lsid:zoobank.org:act:7AE6709F-DECC-4007-8B23-D1292CBBEB07

Ixonema gracielea species LSID: urn:lsid:zoobank.org:act:47A2A602-757A-4C06-A4C3-04540A0B34B7.

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
