# Peer review of "Two new species of Microlaimidae (Nematoda: Microlaimida) from the Continental Shelf off Northeastern Brazil (Atlantic Ocean) with emended diagnosis and dichotomous key"

_PeerJ, doi:10.7717/peerj.17976_

## Round 0.1 · original submission · Minor Revisions

I agree with the reviewers that, in general, the manuscript is well written and the description of the new species is comprehensive. I hope the authors would consider and incorporate the suggestions by all the reviewers and I am looking forward to reading the revised version of this manuscript.

Reviewer 1 ·

Basic reporting

- the text is fairly clear but there are inconsistencies with using either British or American English, and parts of the description and diagnosis are not standardised. Citation of more than 3 authors should be written et al. I would therefore request for the authors to get professional English proofreading.

Literature, tables and figures are sufficient.

Experimental design

For a taxonomic approach its sufficient, just look into the writing

Validity of the findings

impactful since these nematode species are rarely found, and the updates to the keys for the genera are helpful as it covers global scale.

Additional comments

Other than having a novel search, the authors have done well, just to polish their writing

Annotated reviews are not available for download in order to protect the identity of reviewers who chose to remain anonymous.

Reviewer 2 ·

Basic reporting

The authors present the description of two new species of free-living marine nematodes from a poorly known part of the world, Brazilian continental shelf.
The descriptions are very well done and the figures and tables have high quality. I congratulate the authors! I wish that we could have more good nematode descriptions like the ones presented in this manuscript.
I have some suggestion and corrections but they are mostly about the order of information and small text corrections.
For all these reasons I suggest to accept the manuscript for publication after minor corrections.

Experimental design

No comment.

Validity of the findings

No comment.

Additional comments

Here I give my specific comments:

Title: The title could be shorter by omitting some unnecessary details. My suggestion: “Two new species of Microlaimidae (Nematoda: Microlaimida) from the Continental Shelf off Northeastern Brazil with emended diagnosis and dichotomous key”.

Key-words: Please use only words that are not in the Title. If the authors accept my suggestion they can leave “marine nematodes”, otherwise they should substitute it! “Species description” does not say too much… choose more appropriate key word.

Introduction:
Lines 21-32: For me the first paragraph is a little bit confusing. It seems to me that the authors are considering all South Atlantic Ocean as a same region. This is very strange! The Brazilian and Argentinian coasts are very long coastlines with many habitat and climatic variations…they cannot be considered as one region. Please rewrite this paragraph taking this fact in consideration!
Lines 55-57: delete “will” (authors already described species… not future!). Delete “the” before genera names and delete “genus” after genera names… When we see names in italics we already know that they are genera!

Material and Methods:
Table 1 is completely unnecessary. All information of latitude, longitude and depth is in the type and paratype locality parts.
Type locality and Paratype locality: there is no need to present information about station number. The important are only latitude and longitude! Delete this in all locality information through the all manuscript!

Descriptions:
Those not make sense basing description only in holotype (1 male) and 1 female paratype. Species descriptions should be based in all individuals. The author can easily adapt the text considering all individuals. Please, consider this for all descriptions in the manuscript!
Substitute “behind” for “posterior” in all situations through the manuscript (i.e. line 155, 168, 281…)
Write sperm cells, instead of only “sperm”, in all situations through the manuscript (i.e. line 160, 361…figure 4 caption…).
Line 172: check the number of turns in amphidial fovea of females! Looking figure 2 for me is 3 turns! Please consider this information to correct also table, diagnosis and discussion!
Lines 190-191: if the spicules are equal authors should use always spicules in plural when discussing their morphology. Correct in all situations through the manuscript.

Tables: the tables comparing species should be inserted right after the corresponding differential diagnosis part. The table 4 should be after Spirobolbolaimus identification key. The same observation is also valid for table 5, which should be right after Ixonema identification key.

Line 256: “gracielea” is only the species epithet! The species name is “Ixonema gracielea”. Correct it! This is a mandatory correction considering Code of Zoological Nomenclature! Refer to “gracielea” as species epithet!

Lines 341-342: correct to “setae”

Discussion: for me it is very strange this separate discussion part… I do not know if it is a mandatory section in the journal but could be better reallocate the corresponding discussion of each genus right after the differential diagnosis parts.

Line 358: correct to “we included”

Lines 416-417: a reflection to the authors: can we call as a simple assemblage all Microlaimids of South Atlantic? In my opinion we cannot! This reflects also my opinion about the first paragraph of Introduction… I suggest re-writing the last paragraph!

Figure 1 and 2 captions: correct to “sensilla arrangement”

Figure 2: In 2C would be better delete the amphidial fovea in order to have a good view of buccal cavity (similar to 1D!).

Figure 3 caption: delete “arrow indicating” after “posterior end”. It is repeated!

Table 2 and 3: correct to “spicules” (in plural in all situations of the table!)

Table 4 caption: correct to “spicules” (in plural in all situations)

Tables 4 and 5: I noticed that some information discussed in the text are not here in the tables. Include all information about measurements and morphology here in the tables!

Table 5: correct to “amphidial fovea”

·

Basic reporting

The manuscript is well-written, with comprehensive descriptions accompanied by corresponding images. This work provides valuable new morphological observations and contributes to our understanding of the distribution of the family. These insights are crucial for the ecological comprehension of the studied family.

In addition to the description of new species, a significant strength of this work is the morphometric comparison with previously described species, which is effectively presented in tables.

However, the most significant issue identified is the incorrect construction of the species name for Ixonema. To address this, I suggest referring to a recent article by Braby et al., published in May 2024, which provides guidance on how to describe a new species in Zoology to avoid such errors.

Overall, the article requires minimal modification. Nevertheless, I have included my suggestions for improving the clarity of the manuscript in the uploaded PDF.

Experimental design

Not applicable

Validity of the findings

As mentioned on the topic 1: This work provides valuable new morphological observations and contributes to our understanding of the distribution of the family. These insights are crucial for the ecological comprehension of the studied family.

Additional comments

Another issue I found was the drawing of the Spirobolbolaimus female, which does not represents the morphology of the ovaries as described in the text. I suggest to improve the drawing.

---

## Round 0.2 · accepted · Accept

Thank you for addressing all the comments and suggestions from the reviewers. I believe that the final version of this manuscript is ready for publication. Congratulations.